# Up-down Taper Based In-Fiber Mach-Zehnder Interferometer for Liquid Refractive Index Sensing

**DOI:** 10.3390/s19245440

**Published:** 2019-12-10

**Authors:** Xiaopeng Han, Chunyu Liu, Shengxu Jiang, Shuo Leng, Jiuru Yang

**Affiliations:** 1College of Electronics Engineering, Heilongjiang University, Harbin 150080, China; 2171235@s.hlju.edu.cn (X.H.); 2181197@s.hlju.edu.cn (S.J.); 2181200@s.hlju.edu.cn (S.L.); 2Key Lab of Electronics Engineering, College of Heilongjiang Province, Heilongjiang University, Harbin 150080, China; yangjr@hlju.edu.cn

**Keywords:** refractive index, up-down taper, Mach-Zehnder interferometer, intensity demodulation

## Abstract

A novel in-fiber Mach-Zehnder interferometer based on cascaded up-down-taper (UDT) structure is proposed by sandwiching a piece of polarization maintaining fiber between two single-mode fibers (SMF) and by utilizing over-fusion splicing method. The dual up tapers respectively act as fiber splitter/combiner, the down taper acts as an optical attenuator. The structure parameters are analyzed and optimized. A larger interference fringe extinction ratio ~15 dB is obtained to achieve refractive index (RI) sensing based on intensity demodulation. The experimental results show that the RI sensitivity is −310.40 dB/RIU with the linearity is improved to 0.99 in the range of 1.3164–1.3444. The corresponding resolution can reach 3.22 × 10^−5^ RIU, which is 6.8 times higher than wavelength demodulation. The cross sensitivity which caused by temperature fluctuation is less than 1.4 × 10^−4^.

## 1. Introduction

Optical fiber refractive index (RI) sensors are used in biological fields, medicine and environment [1,2,3]. So far, lots of fiber optic RI sensors have been reported in the literature, such as Mach-Zehnder (MZIs) [4,5,6], Fiber Grating [7], Fabry-Pérot (FPIs) [8], Sagnac [9] and multi-mode interferometer [10,11,12]. Among them, Quan [8] prepared an open-cavity FPI by splicing a section of the fused silica fiber tube between photonic crystal fiber (PCF) and single mode fiber (SMF), an ultra-high sensitivity of 30,899 nm/RIU is obtained. Sensors based on surface plasmon resonance can achieve ultra-high sensitivity with a value of 30,000 nm/RIU is obtained [13]. In addition, the above structures are all used wavelength demodulation and they both require optical spectrum analyzer to monitor the spectral shift.

Comparatively, the schemes of intensity demodulation only need a cost-effective power meter, it is convenient in practical applications. Wu constructed a filling-free FPI structure by simply splicing a section of endless multi-mode photonic crystal fiber (MPCF) [14] or single-mode PCF [15] to a conventional SMF, the sensitivities are 21.52 dB/RIU and 52.4 dB/RIU, respectively. Ran [16] etched a short air cavity near the tip of SMF by laser micromachining, and the sensitivity of 27 dB/RIU is obtained. Cascading SMF and multimode fiber structures are proposed in references [17,18], the interference pattern caused by modal interference and the sensitivity of these structures are corresponding to 67.9 dB/RIU and 110 dB/RIU. Furthermore, using SMF splice with a high birefringence fiber [19] or offset-core thin core fiber [20] to form Michelson interference, the maximum sensitivity of the above structure can reach 202.46 dB/RIU. To enhance the sensitivity, some hybrid interference structures consisting of a Bragg grating and other fibers are used for RI testing [21,22,23]. Among them, Zhang [21] composed a SMF-no core fiber-SMF cascaded with two FBGs, the reported sensitivity are 199.6 dB/RIU (in the range of 1.33–1.37) and 355.5 dB/RIU (in the range of 1.37–1.40) by utilizing an intensity difference method. Moreover, tapering is a convenient technical means to improve the RI sensitivity since it can effectively increase the contact area between the in-fiber light power and the external liquid, an enhanced evanescent field as a medium for power exchange at the same time [24,25,26]. Kong [26] designed a hybrid multi-mode interferometer which consisting of a thin core fiber taper and an air bubble fabricated by the arc discharge technique, which exhibits a high RI sensitivity of 442.59 dB/RIU. In addition, using tapered microfiber acts intensity attenuator in a dual-wavelength erbium-doped fiber laser to achieve RI sensing as described in literature [27], a relative high sensitivity of 273.7 dB/RIU is obtained.

In this paper, a compact up-down taper (UDT) structure is proposed in order to realize high sensitivity RI sensing on intensity demodulation. The dual up tapers are act as optical beam splitter and combiner, respectively. The down taper acts as an optical attenuator to monitor the value of the light intensity. Panda-type polarization maintaining fiber (PMF) is used in the sensitive area because it can avoid polarization mode coupling during transmission and effectively reduce the influence of polarization mode dispersion on sensing performance. Additionally, the proposed structure does not require large instrument and complicated manufacturing process, and we can get this structure by fusion splicing. The experiment results show that the up-down taper structure with optimized parameters has higher sensitivity than dual-up-taper structure in RI sensing test. The proposed sensor has a high sensitivity of 310.40 dB/RIU in the range of 1.3164–1.3444 which is more than 6.8 times higher than that in wavelength demodulation. The temperature cross talk is less than 0.014%. Such a structure with the advantages of small size, low cost and less temperature crosstalk can be applied in the fields of high precision detection of RI test.

## 2. Structure and Principles

The structure is illustrated in Figure 1. The length of the PMF is $L_{1}$. A down taper is located at the middle of double up tapers with the length of $L_{2}$, which includes two transition area and a taper waist area. $L_{3}$ is the taper length in the up-taper area. The beam is divided along two propagation paths when the incident light is transmitted to the first up-taper area. One path of light is transmitted in the core of the middle section PMF, the other path is coupled into the cladding of the middle section PMF and transmitted as cladding modes.

The high-order cladding modes would be excited. An optical path difference is generated since the effective refractive index (ERI) of the cladding mode is different from the fiber core mode. In this fiber MZI, dual up tapers respectively act as fiber splitter/combiner, middle section PMF provides effective interference length. The down taper is act as an optical attenuator which can enhance the sensing performance of surrounding environments in the form of evanescent waves. Therefore, corresponding peaks and dips can be obtained in the output spectrum and the output light intensity can be expressed by Equation (1):(1)$$I=I_{1}+I_{2}+2\sqrt{I_{1}I_{2}}\cdot \cos\left[\frac{2\pi \;\Delta n}{\lambda }\cdot L_{1}\right]$$where $\Delta n=n_{eff}^{co}-n_{eff}^{cl}$ is the difference of the ERI between the core and cladding modes, and $I_{1}$ is the power of the core mode, $I_{2}$ is the power of the m-th cladding mode, $L_{1}$ is the length of PMF, λ is the wavelength of incident light, $\Delta \varphi =\frac{2\pi \;\Delta n}{\lambda }\cdot L_{1}$ represents the phase difference between the fiber modes [28]. In Equation (1), the resulting value reaches to the minimum value when the phase difference $\Delta \varphi =\frac{2\pi \;\Delta n}{\lambda }\cdot L_{1}=\left(2m+1\right)\pi \;\left(m\;\mathrm{is}\;\mathrm{an}\;\mathrm{integer}\right)$ [29]. The corresponding transmission spectrum has a depressed peak and can be defined by the following formula: (2)$$\lambda_{dip}=\frac{2\left(n_{eff}^{co}-n_{eff}^{cl}\right)L_{1}}{2m+1}$$

The difference in ERI will change with the external environment perpetuation when the incident light wave is transmitted in the PMF. It will eventually lead the corresponding characteristic wavelength of the dips drift with the change of temperature and RI [30]. Deriving the RI in Equation (1) the following equation can be obtained:(3)$$\frac{dI}{dRI}=\frac{dI_{core}}{dRI}+\frac{dI_{clad}}{dRI}+\frac{{\left(I_{core}I_{clad}\right)}^{-\frac{3}{2}}}{dRI}\cos\left[\frac{2\pi }{\lambda }\left(n_{eff}^{co}-n_{eff}^{cl}\right)L_{1}\right]+\sqrt{I_{core}I_{clad}}\sin\left[\frac{2\pi }{\lambda }\left(n_{eff}^{co}-n_{eff}^{cl}\right)L_{1}\right]\frac{4\pi }{\lambda }L_{1}\frac{dn_{eff}^{cl}}{dRI}$$

According to the Equation (2), it can be seen that the intensity of the interference signal has compound function relationship with the change of RI. The relationship between the ERI and the wavelength of the cladding mode can be expressed as follows:(4)$$\frac{d\lambda_{dip}}{dn}=-\frac{2\pi L_{1}}{\left(2m+1\right)\pi }=\frac{\lambda_{dip}}{n_{eff}^{cl}-n_{eff}^{co}}$$

Under the changed temperature field, the length and optical fiber ERI of the sensor arm can be changed [31,32], so the characteristic wavelength of the dips in the transmission spectrum can be expressed as follows.
(5)$$\frac{d\lambda_{dip}}{dT}\approx \lambda_{dip}\left[\alpha_{core}+\frac{1}{n_{eff}^{co}-n_{eff}^{cl}}\left(\frac{dn_{eff}^{co}}{dT}-\frac{dn_{eff}^{cl}}{dT}\right)\right]=\lambda_{dip}\left[\alpha_{core}+\frac{\xi_{core}n_{eff}^{co}-\xi_{clad}n_{eff}^{cl}}{n_{eff}^{co}-n_{eff}^{cl}}\right]$$where $\alpha_{core}=\frac{1}{L_{1}}\frac{dL_{1}}{dT}$ represents the fiber core thermal expansion coefficient, $\xi_{core}=\frac{1}{n_{eff}^{co}}\frac{dn_{eff}^{co}}{dT}$ and $\xi_{clad}=\frac{1}{n_{eff}^{cl}}\frac{dn_{eff}^{cl}}{dT}$ are represent the thermo-coefficient of the core and cladding, respectively. However, the changed temperature is often used as the main influence factor of the refractive index measurement error, thus we have achieved the RI and temperature sensing experiments independently to evaluate the crosstalk of the RI sensing structure. 

We have simulated the up-taper and down-taper under different geometric parameters. A larger interference fringe extinction ratio is obtained by optimizing the structure parameters. A numerical simulation based on the beam propagating method is used to describe the spatial distribution of light power in the optical fiber. The mesh sizes in the X, Y, and Z directions in the simulation conditions are 0.1 μm, 0.1 μm, and 1 μm, respectively. The boundary condition of the model is set to the perfect matching layer (PML) condition. The beam behavior of the MZI interference structure based on dual-up-taper (DUT) and UDT are simulated as shown in Figure 2.

The incident light center wavelength is 1550 nm. Assuming that $L_{1}=2.50$ cm, the diameter of PMF is 7.0/125 μm, $n_{co}$ is 1.4565 and $n_{cl}$ is 1.4378. The diameter of SMF is 8.3/125 μm, $n_{co}$ is 1.4565 and $n_{cl}$ is 1.4468. The taper length $L_{3}$ in the up-taper area is 240μm, simulated environmental medium $n_{0}$ is 1.0. For DUT structure, the relationship between outer diameter of up taper (expressed by $d_{1}$) and fringe extinction ratio are shown in Figure 2a. It is clear that the maximum fringe extinction ratio can reach 12 dB when $d_{1}$ is 205 μm. The fringe extinction ratio is increased as the fiber diameter increases in the range of $d_{1}$ < 205 μm. Conversely, the fringe extinction ratio is decreased as the fiber diameter increases in the range of fiber $d_{1}$ > 205 μm. When $d_{1}$ is 205 μm, the structure of UDT is simulated. The relationship between different shortest diameter of down taper (expressed by $d_{2}$) and fringe extinction ratio are shown in Figure 2b. When $d_{2}$ is 30 μm, the maximum fringe extinction ratio can reach 15.5 dB. The maximum fringe extinction ratio is decreased as the fiber diameter increases in the range of 30 μm < $d_{2}$ < 50 μm. On the contrary, the maximum fringe extinction ratio tends to be increased with the increase of down taper diameter in the range of 10 μm < $d_{2}$ < 30 μm. According to the above results, we could construct an up-down taper structure which the outer diameter of the up taper $d_{1}$ is 205 μm and a down taper with diameter $d_{2}$ of 30 μm in order to get a large fringe extinction ratio.

An approximated beam propagation method is used to analyze the electric field distribution, the power fluctuation of each fiber mode is calculated with the propagation distance. The simulated parameters that have been used as follow: the length of PMF is 2.50 cm with a core refractive index 1.4565 and 1.4378 in the cladding. The length of SMF is 1.00 cm, the refractive index of the core and cladding are 1.4565 and 1.4468, respectively. For the up-taper region, the maximum outer diameter and the taper length are 205 μm and 240 μm, respectively. The length of the down-taper is 310 μm, which corresponding the shortest diameter is 30 μm. The input radial field distribution is assumed to be Gaussian distribution, and the fundamental mode of the input SMF is symmetric about the radial circle.

Figure 3a,c show the field amplitude distributions of the transmitted light for the DUT and UDT structures, respectively. It is clearly observed that a small part of the light power is leaked into the cladding area when the light propagates through the taper area. The normalized values of the fiber modes power for DUT structure are shown in Figure 3b. The transmission loss is close to zero in the lead-in SMF portion. When the light is transmitted to the first up-taper area, a part of the light power leaks into the cladding and excites the cladding modes. In the second up taper area, the light power re-coupled into the core, while a small portion of the light power still in the cladding area.

Figure 3d depicts the distribution of normalized fiber modes power for UDT structure. The down-taper is located at z = 20,000 μm in the simulation structure. It is noticed that partial light power in the core is further leaked out at the down taper area for the UDT structure.

Compared with Figure 3b, the number of the cladding modes are increased. Figure 4 shows the normalized power of each $LP_{mn}$ mode when crossing in the DUT and UDT structures. For the DUT structure as shown in Figure 4a it can be seen that 70.70% of the total power still exists in $LP_{01}$ mode, $LP_{02}$ mode carries about 28.16% of total power and $LP_{21}$ mode only occupies 1.14% of the total power.

For the UDT structure as shown in Figure 4b, the results show that the mainly power of $LP_{01}$ mode is reduced to 64.89% of the total power because of the down-taper exist. Meanwhile, the $LP_{11}$ mode and $LP_{12}$ mode are excited. The $LP_{11}$ mode occupies a major light power, which approximately accounts 22.41% of the total energy, the proportion of $LP_{02}$ mode is reduced to 7.31% of total power for the existence of multiple modes. $LP_{21}$ and $LP_{12}$ modes occupy a small proportion of the total power (5.03% and 0.36%). The transmission spectrum is not completely symmetrical since more than two modes are involved in the interference. We can assume that $LP_{11}$ mode is dominantly excited in UDT structure, and the other high-order modes can also modulate the interference mode pattern, but the modulation effect is very weak. Compared with DUT, the ratio of cladding and core power in the structure is increased from 41.44% to 54.11%. Further, the UDT structure can improve the RI sensitivity on the intensity demodulation according to Equation (3). 

## 3. Experiments

A part of uncoated SMF (SMF-28, Corning, New York, NY, USA) is over-fusion spliced with a piece of the PMF (PM15-1(06002)-3) by a commercial fusion splicer (FSM-100P, Fujikura, Tokyo, Japan). The splicing parameters are set up as follows: the overlap size is 100 μm, the arc discharge intensity is 400 bits and the taper discharge time is 150 ms. Secondly, another up taper is made in the same way as above. The middle of the PMF is positioned and tapered by arc discharge method, the arc parameters are set up as follows: the length of taper area is set as 310 μm, discharge intensity is 600 bit and the taper discharge time is 60 ms. The microscope images of down-taper and UDT structure are shown in Figure 5a,b, respectively. It can be clearly seen that a down-taper is located in the middle of the dual up tapers. For the down taper, the symmetric transitions are demonstrated with the length of $L_{2}=310\;\mu m$, the shortest waist-diameter is $d_{2}$= 30 μm. The outer diameters of the up taper $d_{1}$ and taper length $L_{1}$ are measured around 203 μm and 241.8 μm, respectively.

The experimental set up is shown in Figure 5c. The input source is a broadband source (BBS, with the output optical power is 120 mW, wavelength range from 1525–1565 nm), the output interference spectrum is detected by the optical spectrum analyzer (OSA, mod. 86142B, Agilent, Palo Alto, CA, USA) with a resolution of 0.06 nm/0.01 dB). A polarization controller (PC) is used to adjust the polarization states of the input light, which can compensate for any change of polarization state induced by fiber loops and twists in the optical path leading to the PMF. When the incident light is propagated into the first-stage up-taper region, the beam is split into two parts. One part of the beam continues to propagate in the core as $LP_{01}$ mode, another part propagating in the cladding region mainly as $LP_{02}$ mode. When the incident light is reach to the down-taper region, the light power in the core is further diffused into the cladding caused by the enhanced evanescent field, and mainly propagated in the form of $LP_{11}$ mode. Meanwhile, the value of light power in the cladding region has increased. A part of the light power in the cladding is recoupled back into the core at the second-stage up-taper region. The mainly mode interference phenomenon between the cladding mode $LP_{02},$
$LP_{11}$ and the core mode $LP_{01}$ are generated. We designed two different structures, which are UDT and DUT, respectively. For the DUT structure, the external environment mainly affects the change of the ERI of the $LP_{02}$ mode in the cladding. As to the UDT structure, the change of the RI in the external environment mainly affects ERI of the $LP_{11}$ mode in the cladding. This is also the reason why the interference phenomena of the two structures are different. The change in the RI of the external environment can lead to the variety of power by influencing the cladding mode. The characteristic wavelength and intensity corresponding to the dips of interference are drift in different degree. In the experiment, the mixed solvent of water and glycerin is used to achieve RI sensing, we have prepared different concentrations (0%–20.99%) of glycerol solution and calculated the corresponding refractive index by the following equation [33]: n=1.33303+[0.0011489×c+S], w here c represents the concentration, and S represents the specific gravity (S=1.26331 at room temperature 25 °C). An Abbe-refractometer (RI monitoring range is 1.3000–1.7000, working wavelength 589 nm) is used for the RI test. We calibrate the RI values obtained from the experiment to the value corresponding in 1550 nm band [34].

Three samples with lengths $L_{1}$ = 1.0 cm, 1.5 cm, and 2.5 cm, respectively, are prepared as DUT structures for comparison. The results show that the distribution of the transmission spectrum is relatively uniform when $L_{1}=2.5\;\mathrm{cm}$. Using the fast Fourier transform (FFT) method to transfer the data of the spectrum, the relationship between the spatial frequency ξ and the difference of ERI can be given as $\xi =\frac{1}{\lambda^{2}}\cdot \Delta n_{eff}\cdot L_{1}$, which indicates that the difference of ERI is proportional to spatial frequency [35]. The results are displayed in Figure 6a,b.

A dominant cladding mode and core mode are formed interference which can be considered. Such as the ones illustrated in Figure 6a, the transmission spectrum is formed after over-fusion splicing of PMF ($L_{1}$ = 2.5 cm) and SMF. Three dips are formed in the spectrum, in detail, the free spectral range of two adjacent dips is 12.74 nm and 12.24 nm, respectively. The maximum extinction ratio of the interference fringes is about 12.677 dB. It can be seen from the FFT analysis in Figure 6a that there are a few peaks in the spatial spectrum, indicating that multiple modes are involved in the interference and the dominant cladding mode $LP_{02}$ which corresponding frequency is 0.089 (1/nm). The transmission spectrum of the UDT structure and corresponding FFT spectrum are shown in Figure 6b. Compared to DUT structure, the number of individual cladding modes has changed obviously, and the corresponding frequency of dominant cladding mode $LP_{11}$ is 0.044 (1/nm). Since the power of the core mode is further diffused into the cladding at the down taper area, the power corresponding to the mainly interference cladding modes have increased, the interference fringe extinction ratio of the two dips can reach to 14.903 dB and 14.037 dB, respectively. The UDT structure can be used to further increase the interference between core mode and dominant cladding modes.

The experiment is conducted at room temperature (25 ± 0.2 °C). A comparison RI sensing experiment using DUT structure is carried out. The evolution of the transmission is shown in Figure 7a, the wavelength of dip2 shows blue shift when the RI increases and the intensity of fringe is increased. The wavelength of dip2 shifts against the RI and linear response are plotted as shown in Figure 7b, the experiment results show that a sensitivity of 168.73 dB/RIU when the liquid RI corresponds in the range of 1.3164–1.3473. However, the wavelength is only blue shifts by 0.405 nm in this range. The transmission spectrums of UDT structure with the varied surrounding RI are shown in Figure 7c, we can see that as the RI increases, the wavelength of dip2 shows blue shift and the intensity of fringe is decreased. The wavelength of dip2 shifts against the RI is plotted and linearly fitted in Figure 7d, the results show that a sensitivity of 310.40 dB/RIU from the resonance wavelength when the liquid RI corresponds in the range of 1.3320–1.3605. The wavelength did not change significantly and it drifts only ~1.035 nm in the range of the RI interval. The detection resolution is 6.8 times to the wavelength demodulation by using intensity demodulation. Compared with the RI experiment results of the DUT structure which are shown in Figure 7a, the RI sensitivity of the UDT structure on the intensity demodulation is increased 83.96% with better linearity. 

The change of ambient temperature can be considered as measuring RI errors, and the temperature sensitivity is also measured within the range of 30–65 °C. As shown in Figure 8a, we can see that the wavelength of dip shows a red shift as the temperature increases, and meanwhile the intensity of fringe is increased. A sensitivity of 0.20970 dB/°C and 41.21 pm/°C in the range of 30–65 °C as presented in Figure 8b. The resolution of the sensor with the RI sensitivity is 3.22 × 10^−5^ RIU. The experimental measurement error is 0.01351% which is 0.04194 dB/RIU light intensity crosstalk in cross sensing experiment for the environment temperature change is ±0.2 °C. Based on the above experimental results, the proposed sensing structure has different sensitivity to RI and temperature. The change in wavelength drift and light intensity can be expressed as the following equations [36]:(6)$$\{\begin{array}{l} \Delta \lambda =K_{n\lambda }\Delta n+K_{T\lambda }\Delta T\\ \Delta I=K_{nI}\Delta n+K_{TI}\Delta T \end{array}$$where $K_{n\lambda }$ and $K_{T\lambda }$ are the sensitivities of the wavelength shift corresponding to RI and temperature, $K_{nI}$ and $K_{TI}$ are the sensitivities of the output light intensity corresponding to RI and temperature, respectively. Equation (6) can be expressed in the following demodulation matrix:(7)$$\left[\begin{array}{c} \Delta \lambda \\ \Delta Ι \end{array}\right]=\left[\begin{array}{cc} K_{n\lambda } & K_{T\lambda }\\ K_{nI} & K_{TI} \end{array}\right]\left[\begin{array}{c} \Delta n\\ \Delta T \end{array}\right]$$

All the sensitivity coefficients can be used as the sensitivity matrix to obtain the variation in RI and temperature:(8)$$\left[\begin{array}{c} \Delta n\\ \Delta T \end{array}\right]=\frac{1}{D}\left[\begin{array}{cc} K_{TI} & -K_{T\lambda }\\ -K_{nI} & K_{n\lambda } \end{array}\right]\left[\begin{array}{c} \Delta \lambda \\ \Delta Ι \end{array}\right]$$where $D=K_{TI}K_{n\lambda }-K_{nI}K_{T\lambda }$, substitute the experimental results into above formula and a demodulation matrix is established, which is expressed as:(9)$$\left[\begin{array}{c} \Delta n\\ \Delta T \end{array}\right]=\frac{1}{-19.975}\left[\begin{array}{cc} 0.20970 & -0.04121\\ -310.40 & -34.255 \end{array}\right]\left[\begin{array}{c} \Delta \lambda \\ \Delta Ι \end{array}\right]$$

Therefore, according to the light intensity changes with RI and temperature, the RI sensitivity of the proposed sensor is −310.40 dB/RIU, and the sensitivity of temperature in intensity modulated is only −0.20970 dB/°C. The proposed RI sensing structure has lower temperature crosstalk with a cross-sensitivity is 0.00068 RIU/°C.

Table 1 compares other sensors with different structures. The resolution of the OSA used in the above literature is 0.01 dB and the measurement resolution is also calculated. By comparison, we can find that the RI sensitivity based on intensity demodulation and detection resolution of the structures in this paper is obviously higher than other structures. 

## 4. Conclusions

In this paper, a fiber optic measurement of RI sensor with UDT structure is designed. The structure can be used as an all-fiber type device integrating double optical beam splitters/combiners and an optical attenuator. Dual up tapers act as fiber splitter and combiner, respectively. The down taper acts as an optical attenuator to monitor the output light intensity. We found that the structure based on MZI principle has been improved the performance significantly compared to the DUT structure. Through simulation analysis, the main reason can be interpreted as the fact the down taper causes the weaker evanescent field which is located between two up tapers to be enhanced. At the same time, the optimized sensing structure has a large fringe extinction ratio reaching 14.903 dB. The proposed structure has certain advantages in intensity demodulation for RI sensing since the core power and the cladding power are approximately equally distributed. The experimental results show that the RI sensitivity is 310.40 dB/RIU with a linearity of 0.99 in the range of 1.3164–1.3444, and the error from cross-sensitivity is less than 1.4 × 10^−4^ owing to the crosstalk for temperature change. Such a UDT sensing structure has a good application prospect in the field of high precision RI detection tests.

## Figures and Tables

**Figure 1 sensors-19-05440-f001:**
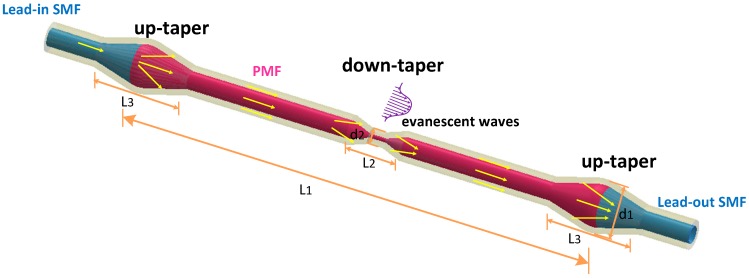
Up-down taper structure.

**Figure 2 sensors-19-05440-f002:**
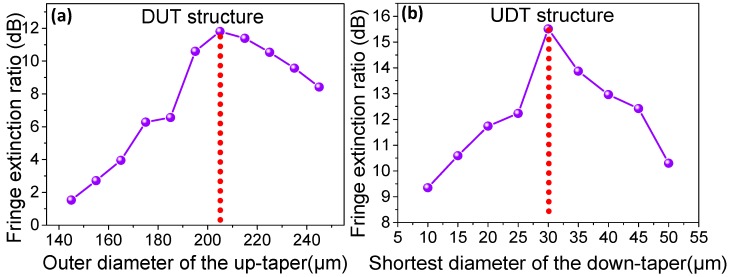
The relationship between (**a**) the diameter of the up taper and spectral fringe extinction ratio for DUT structure; (**b**) the diameter of the down taper and spectral fringe extinction ratio for UDT structure.

**Figure 3 sensors-19-05440-f003:**
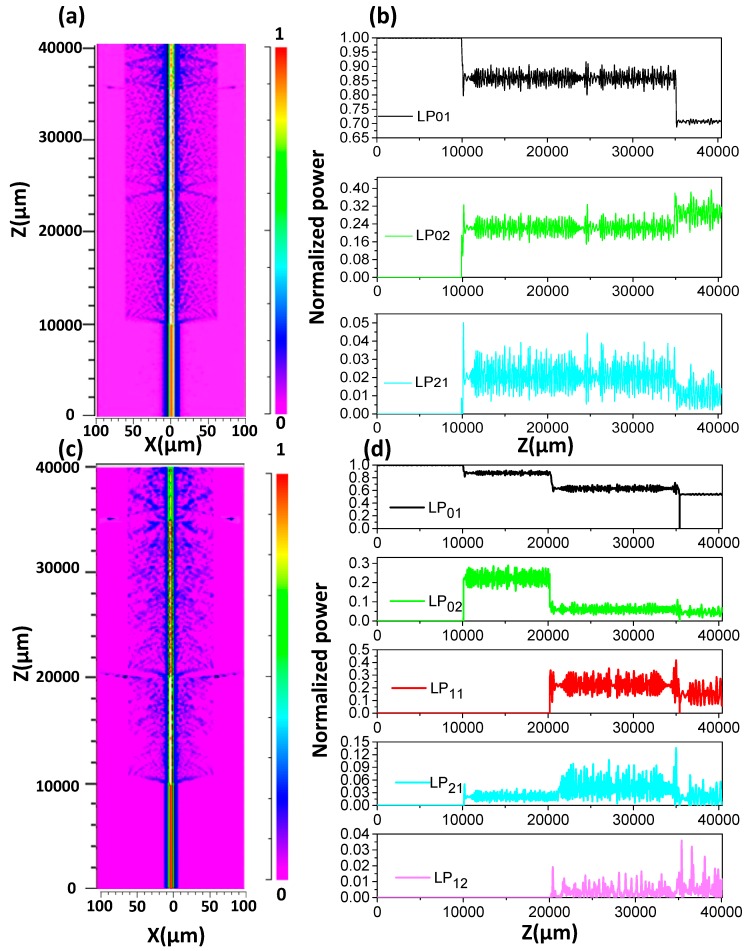
(**a**) Simulated optical field power within the DUT structure; (**b**) Power fluctuation of each mode with transmission distance Z in the DUT; (**c**) Simulated optical field power within the UDT structure; (**d**) Power fluctuation of each mode with transmission distance Z in the UDT.

**Figure 4 sensors-19-05440-f004:**
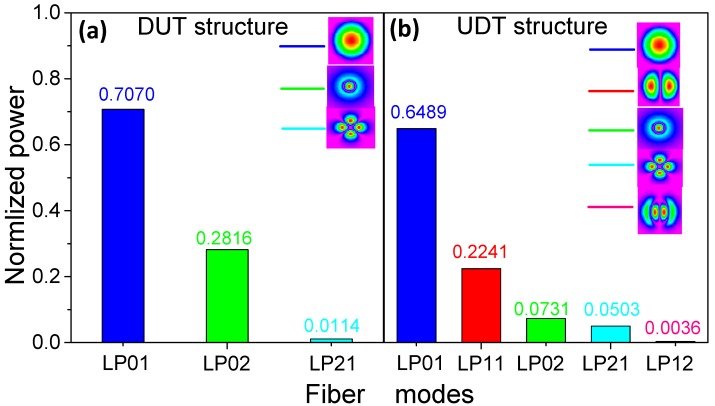
The normalized power of each $LP_{mn}$ mode crossing in the (**a**) DUT and (**b**) UDT structure.

**Figure 5 sensors-19-05440-f005:**
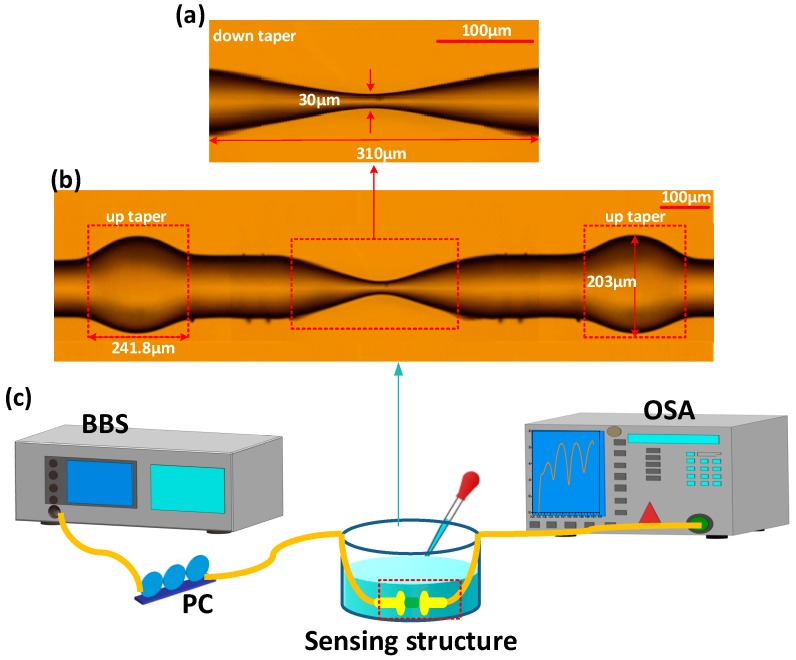
Microscope images of (**a**) down-taper and (**b**) the UDT structure; (**c**) The experimental set up.

**Figure 6 sensors-19-05440-f006:**
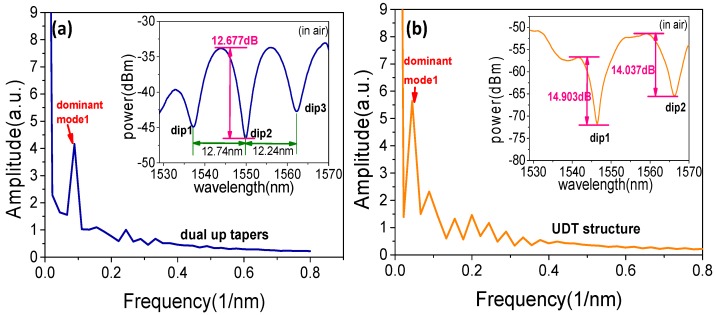
The transmission spectrum and Fourier spectrum analysis of the (**a**) DUT and **(b**) UDT structures.

**Figure 7 sensors-19-05440-f007:**
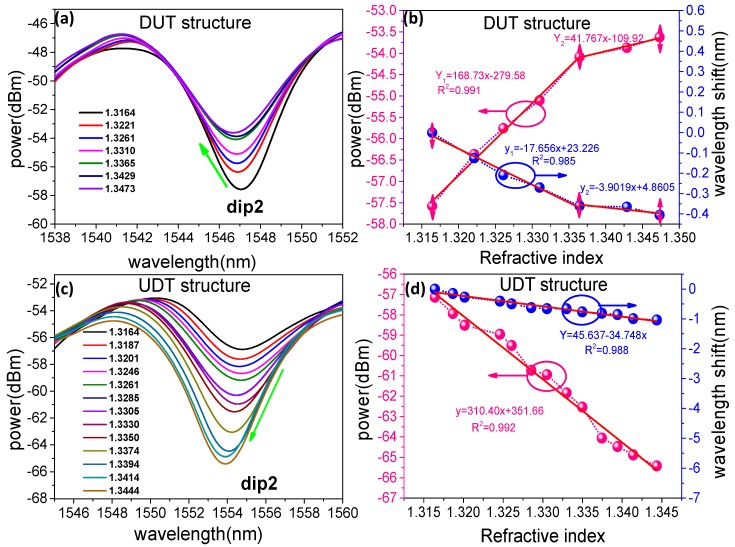
(**a**) RI measured transmission spectra of the DUT structure; (**b**) Experiment results of fringe power and wavelength as a function of RI for a dual up tapers structure; (**c**) Refractive index measured transmission spectra of the up-down taper structure; (**d**) Experiment results of fringe power and wavelength as a function of RI for up-down taper structure.

**Figure 8 sensors-19-05440-f008:**
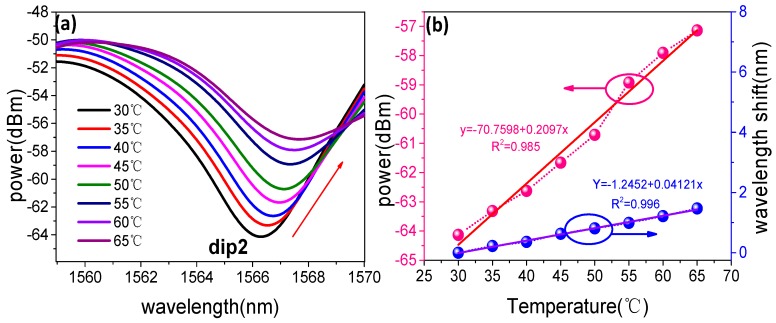
(**a**) Temperature measured transmission spectra of UDT structure; (**b**) Experiment results of fringe power and wavelength as a function of temperature for UDT structure.

**Table 1 sensors-19-05440-t001:** Comparisons to different structure and their performance.

Sensing Structure	RI Sensitivities	Test Range	RI Resolution	Ref
**Fabry-Pérot** **interferometer**	21.52 dB/RIU	1.332–1.440	4.65 × 10^−5^ RIU	[14]
**Fabry-Pérot** **interferometer**	52.4 dB/RIU	1.33–1.43	1.91 × 10^−4^ RIU	[15]
**SMS**	−67.9 dB/RIU	1.33–1.3737	1.47 × 10^−4^ RIU	[17]
**Michelson** **Interferometer**	30.1141dB/RIU	1.3337–1.4275	3.32 × 10^−4^ RIU	[19]
**SNS structure** **Cascaded two** **FBGs**	−355.5 dB/RIU −199.6 dB/RIU	1.3702–1.4066 1.3326–1.3702	2.81 × 10^−5^ RIU 5.01× 10^−5^ RIU	[21]
**Multimode microfiber**	−72.247 dB/RIU	1.332–1.355	1.38 × 10^−4^ RIU	[24]
**Erbium-doped fiber laser**	−273.7 dB/RIU	1.300–1.335	3.65 × 10^−5^ RIU	[27]
**Up-down taper**	−310.40 dB/RIU	1.3164–1.3444	3.22 × 10^−5^ RIU	Our work

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
