# Peer review of "Up-down Taper Based In-Fiber Mach-Zehnder Interferometer for Liquid Refractive Index Sensing"

_sensors, 2019, doi:10.3390/s19245440_

Round 1
Reviewer 1 Report
The authors designed an ambient refractive index sensor with a down-taper sandwiched by two up-tapers. The authors give result of their theoretical explanation, numerical simulation, and the experiments of the sensor. This intensity-based sensor shows a high refractive index sensitivity. Still, we have some question to discuss with the authors:
In the introduction part, the authors stated that “However, the RI sensitivity of these sensing structures is lower than 10000nm/RIU.” But a sensitivity of 30899 nm/RIU has already been achieved using a Fabry-Pérot cavity based optical fiber sensor. In the introduction part, the authors stated that “But they are susceptible to external environment perturbation such as temperature.” But there are a lot of publications propose surface plasmon resonance-based sensor which measure both temperature and refractive index. In the first paragraph of section “2. Structure and Principles”, the abbreviation of “ERI” should be explained when it appears for the first time. In the paragraph below Eq. (1), “In Equation (1), the resulting value reaches to the minimum value when the phase difference [equation] (k is an integer) [29].”, it’s hard to understand when the phase difference equals to the expression given by the authors the interference power will have a minimum value since the term cos[expression] may not be -1. Moreover, k always stands for wavenumber in optics, it will be better to use another character if it’s just an integer in this article. Substitute Eq. (2) into Eq. (1), the third term in Eq. (1) becomes 2*sqrt(I1I2)*cos(2*k + 1), k is an integer. This may not be the minimum value, therefore the expression given in the right side of Eq. (2) may not be the “depressed peak”. Please check. Please use the same symbols for the same variables. In Eq. (3), according to the expression given by the authors, Iclad is considered as a function of RI. If so, seems the authors lost some terms in this calculation. Please check. Under Eq. (3), the authors state that “And it can be seen that the intensity of the interference signal has a linear relationship with the change of RI.” Seems the right side of Eq. (3) is not a constant, please check. More explanation of Eq. (5) is necessary, seems it’s not readily available from Eq. (2). An clarify of the term “thermo-coefficient” used in this article is necessary (below Eq. (5)). In the paragraph below Fig. 2, the authors state that “The fringe extinction ratio is decreased as the fiber diameter increases in the range of d1<205μ” But according to Fig. 2, seems the fringe extinction ratio increases with the increase of the fiber diameter. In the paragraph below Fig. 4, the authors state “It is clearly observed that most of the light power is leaked into the cladding area when the light propagates through the taper area.” Seems most of the power still constrained inside the core according to Fig. 3. In the paragraph below Fig. 4, the authors state that “It is noticed that more light power in the … are increased simultaneously.” It is hard to know that the cladding modes are increased from Fig. 3 (d). In the paragraph above Fig. 5, it could be better if the authors clarify if the light source used in this experiment is a polarized light source since the authors used a polarization controller to control the polarization of the incident light. It is necessary for the authors to clarify the wavelength of the Abbe-refractometer used in this experiment. In the paragraph below Fig. 6, the authors state that “It can be seen from the FFT analysis in … corresponding frequency is 0.089 (1/nm).” It is hard to understand there's a relationship between the spatial frequency and the fiber mode. In Table 1, seems the sensor proposed by Ref. [13] is not a Fabry-Pérot interferometer.
Author Response
Thank you very much for your letter and reviewer's comments on our manuscript entitled " Up-down taper based in-fiber Mach-Zehnder Interferometer for liquid refractive index sensing" (ID:651939). Your opinion is very helpful for revising and improving our paper. At the same time, your suggestions have provided us with better research ideas, and make our research papers more rigorous. We have carefully studied the comments and corrected them. And we are hoping to get your approval. The outline of the issues you proposed is marked in red and the corresponding modified content is marked in black in the text. The blue note is the revised content in the article. The main corrections in this article and the responses to the reviewers' comments are shown in the word attachment below.

Reviewer 2 Report
Several mistakes appear in the reference making it hard to be understood. For example, fiber sensors in Ref 1-3 are not employed in RI sensing as mention in the first line of the introduction. Ref 7 is not a MZI structure. Ref 10 and Ref 11 are the same. THe MPCF should be Ref 14. REf 21 is the structure of SMF -No core fiber-SMF. Several mistakes in the manuscript can be found, such as ERI is not defined as it is used for the first time, there is no Δφ in Eq. (1), and the description for Δφ is also incorrect.... It is hard to see the linear relationship between intensity variation and the index change as mentioned by the authors in Eq. (3). The description for Fig 2(a) as d1<205um is wrong. In fig 3(c), why I can only see field discontinuity in the beginning of down taper? There should be another discontinuity appears int the end of the down taper. Besides, why the power is increased after the down taper as shown in Fig. 3(c)? The down taper region in the proposed structure can be considered as another coupler, and the whole structure can be regarded as cascaded MZIs, which also results in higher ER and sensing sensitivity as presented in [L. M. Hu, C. C. Chan, X. Y. Dong, Y. P. Wang, P. Zu, W. C. Wong, W. W. Qian, and T. Li, “Photonic crystal fiber strain sensor based on modified mach-zehnder interferometer,” IEEE Photonics J., vol. 4, no. 1, pp. 114–118, 2012.] Please verify the difference in ER-enhanced mechanism for the proposed structure and the reference structure. In Fig. 7(b), the sensing sensitivity is reduced for larger RI, which is quite different to other structures. Why? How to obtain the temperature cross sensitivity of 0.00014 should be explained. Table I does not contain the results of Ref 21 which has higher RI sensitivity as mentioned in Introduction.
Author Response
Thank you very much for your letter and reviewer's comments on our manuscript entitled " Up-down taper based in-fiber Mach-Zehnder Interferometer for liquid refractive index sensing" (ID:651939). Your comments are invaluable and are very helpful in revising and perfecting our papers. And it has important guiding significance for our research. We have carefully studied the comments and corrected them. We hoping to get your approval. The outline of the problem you proposed is marked in red and the corresponding modified content is marked in black in the text. The blue font is the modified content in the article.The main corrections in this article and the responses to the reviewers' comments are shown in the word attachment below.

Reviewer 3 Report
The authors report a RI sensor based on an in-fiber type Mach-Zehnder Interferometer by simulations and experiments. They compared their sensor with the recent works on RI sensors based on several interferometers and optical fibers in Table 1 which looks a nice summary for the recent RI sensors. The manuscript will be suitable for Sensors after appropriate modification as follows.
[English style and small literal errors]
The conjunction word "And" frequently appeared at the beginning of a sentence. The authors should replace it into appropriate conjunction or modify the sentence. At p.1, L.2 in Introduction, the word "literatures" should be "literature". In general, "literature" is a non-count noun. At p.3, L.6, the abbreviated term "ERI" appeared but the full spelling was given at L.13. At p.7, L. 3, the commercial name "Fujikua" is "Fujikura"? At p.7, L.14, "120mw" should be "120mW"?
[Simulation]
No simulation condition (approximated beam propagation method) was described.
[Experiment]
a) The authors used glycerol as a solvent. However, the RI dependence was investigated (Fig. 7) which might be achieved by the mixed solvent of water and glycerol. The authors should describe the experimental condition. b) The authors represented the analytical sensitivity with 7 digits e.g. 171.7641 and 305.7477 dB/RIU although the RI of the solvent has 5 digits. The authors should treat the results with the appropriate significant digits less than 5 digits. The Mach-Zehnder interferometer here was applied at around 1550 nm in this study. The RI of the medium was measured by Abbe-refractometer maybe at visible wavelength. The authors should describe the condition of the RI measurement. The analytical RI sensitivity was evaluated although the RI values at 1550 nm might be different from them. Indeed, the Mach-Zehnder interferometer can detect RI difference but might not detect the RI itself measured by Abbe-refractometer. Therefore, I wonder about the so much significant digits of RI sensitivity and RI values.
Author Response
Thank you very much for your letter and reviewer's comments on our manuscript entitled " Up-down taper based in-fiber Mach-Zehnder Interferometer for liquid refractive index sensing" (ID:651939). Your comments are invaluable and are very helpful in revising and perfecting our papers. It has important guiding significance for our research. Your rigorous research spirit is worthy of our study. We have carefully studied the comments and corrected them. We hope to get your approval. The outline of the problem you are proposed is marked in red and the corresponding modified content is marked in black in the text. The blue font is the revised content in the manuscript.The main corrections in this article and the responses to the reviewers' comments are shown in the word attachment below.

Reviewer 4 Report
Please carefully revise the English grammar and the typo errors of the manuscript.
Authors mention that their device can be applied for humidity, strain, and curvature test. However, no results were presented related to these applications. Please, add experiments or remove this sentence.
Please explain why the MZI was employed to the proposed device. Is this the best interferometer for RI measurements? Any other interferometer was tested?
I suggest to write: “Mach-Zehnder (MZIs)[4-7], Fabry-Pérot (FPIs)[8], Sagnac [9] and multi-mode interferometer”
Please check if it’s right: “The dual up tapers are respectively acts as fiber splitter,”.
Was the Up-down taper structure showed in Figure 2 a commercial or a lab made device? Please improve experimental details.
How the simulations of Figure 3 were performed? What parameters or software were employed in these simulations? It’s difficult to assess the results without this information, please improve it.
Please explain why the normalized power is analyzed instead of the intensity in Figures 3b and 3d.
Figure 4 must be corrected.
Improve the legend of Figure 5.
I suggest correcting: Fig.6. The transmission spectrum and Fourier spectrum analysis of the (a) DUT and (b) UDT structures.
Please improve the description of how the refractive index were varied in experiments. Which liquids were evaluated for the proposed device?
Author Response
Thank you very much for your letter and reviewer's comments on our manuscript entitled " Up-down taper based in-fiber Mach-Zehnder Interferometer for liquid refractive index sensing" (ID:651939). Your comments are invaluable and are very helpful in revising and perfecting our papers. And it has important guiding significance for our research. We have carefully studied the comments and corrected them. We hope to get your approval. The outline of the problem you are proposed is marked in red and the corresponding modified content is marked in black in the text. The blue note is the revised content in the article.The main corrections in this article and the responses to the reviewers' comments are shown in the word attachment below.

Round 2
Reviewer 1 Report
The authors have addressed reviewers' comments successfully. Recommended for publication.
Author Response
Thank you very much again for your letter and reviewer's comments on our manuscript entitled " Up-down taper based in-fiber Mach-Zehnder Interferometer for liquid refractive index sensing" (ID:651939). Your comments are very valuable and make our paper more rigorous. Each of your suggestions is better to guide our next research work and further improved our scientific research capabilities. At the same time, it has important guiding significance for our research. At last, we would like to express our gratitude towards your careful review.
Reviewer 2 Report
on line26, in Ref8, the FP fiber sensor is formed by splicing a HCF between a AMF and a PCF, not just splicing a PCF with a SMF. why use 0.2oC as temperature variation? The sensitivity variation of Fig. 7(b) may resulted in different cladding modes were induced. Do the results in Fig. 4. change as the RI of fiber outside is varied? How the down taper affects the RI sensitivity? How to explain the FSR variations in Fig. 6(a) and 6(b)?
Author Response
Thank you very much again for your letter and reviewer's comments on our manuscript entitled " Up-down taper based in-fiber Mach-Zehnder Interferometer for liquid refractive index sensing" (ID:651939). Your comments are very valuable and make our paper more rigorous. Each of your suggestions is better to guide our next research work and further improved our scientific research capabilities. At the same time, it has important guiding significance for our research. At last, we would like to express our gratitude towards your careful review. We have carefully studied the comments and corrected them, hoping to get your approval. The outline of the problems you are proposing is marked in red and the corresponding modified content is marked in black in the text. The green note is the revised content in the article. The main corrections in this article and the responses to the reviewers' comments are as follow attachment.

Reviewer 4 Report
The authors have corrected all suggested recommendations and the article is now ready to be published.
Author Response

(The authors gave the same response as above.)
